# Large-Scale Fabrication of Tunable Sandwich-Structured Silver Nanowires and Aramid Nanofiber Films for Exceptional Electromagnetic Interference (EMI) Shielding

**DOI:** 10.3390/polym16010061

**Published:** 2023-12-23

**Authors:** Xinbo Jiang, Guoqiang Cai, Jiangxiao Song, Yan Zhang, Bin Yu, Shimin Zhai, Kai Chen, Hao Zhang, Yihao Yu, Dongming Qi

**Affiliations:** 1Key Laboratory of Advanced Textile Materials and Manufacturing Technology and Engineering Research Center for Eco-Dyeing & Finishing of Textiles, Zhejiang Sci-Tech University, Hangzhou 310018, China; jxb1222@outlook.com (X.J.); ssongjx0721@163.com (J.S.); zsm021616@163.com (S.Z.); chenkai@zstu.edu.cn (K.C.); 18868166936@163.com (H.Z.); 2Nice Zhejiang Technology Co., Ltd., Hangzhou 310018, China; caigq84@cnnice.com; 3Key Laboratory of Green Cleaning Technology & Detergent of Zhejiang Province, Lishui 323000, China; 4Shaoxing-Keqiao Institute, Zhejiang Sci-Tech University, Shaoxing 312000, China; 5State Key Laboratory of Fire Science, University of Science and Technology of China, 96 Jinzhai Road, Hefei 230026, China; yubin@ustc.edu.cn; 6Zhejiang King Label Technology Co., Ltd., Huzhou 313100, China; yyh@king-grp.com

**Keywords:** EMI shielding, sandwich structure, aramid nanofibers, AgNWs

## Abstract

The recent advancements in communication technology have facilitated the widespread deployment of electronic communication equipment globally, resulting in the pervasive presence of electromagnetic pollution. Consequently, there is an urgent necessity to develop a thin, lightweight, efficient, and durable electromagnetic interference (EMI) shielding material capable of withstanding severe environmental conditions. In this paper, we propose an innovative and scalable method for preparing EMI shielding films with a tunable sandwich structure. The film possesses a nylon mesh (NM) backbone, with AgNWs serving as the shielding coating and aramid nanofibers (ANFs) acting as the cladding layer. The prepared film was thin and flexible, with a thickness of only 0.13 mm. AgNWs can easily form a conductive network structure, and when the minimum addition amount was 0.2 mg/cm^2^, the EMI SE value reached 28.7 dB, effectively shielding 99.884% of electromagnetic waves and thereby meeting the commercial shielding requirement of 20 dB. With an increase in dosage up to 1.0 mg/cm^2^, the EMI SE value further improved to reach 50.6 dB. The NAAANF film demonstrated remarkable robustness in the face of complex usage environments as a result of the outstanding thermal, acid, and alkali resistance properties of aramid fibers. Such a thin, efficient, and environmentally resistant EMI shielding film provided new ideas for the broad EMI shielding market.

## 1. Introduction

With the rapid development of communication technology and the widespread deployment of public mobile communication base stations, electronic signals now cover the vast majority of areas [1,2,3,4,5]. However, the resulting EM pollution brings about a host of problems that seriously affect both human health and the normal operation of high-precision electronic equipment [6,7,8]. The development of flexible electronic devices has led to an increasing demand for the application of electromagnetic shielding films, and thus it is important to develop high-performance electromagnetic shielding films [9,10,11,12]. Nevertheless, current challenges faced by EMI shielding films also include low shielding efficiency, inadequate mechanical properties, and difficulties in large-scale production, which significantly restrict their application and development [13,14,15]. Aiming to alleviate these issues, a fast and effective laminating film is being studied.

The thickness of EMI shielding materials is positively correlated with their shielding properties, making the improvement of an EMI shielding film’s efficiency a key focus in research [16,17,18,19]. In order to address this issue, researchers have conducted a diverse range of investigations, primarily focusing on the utilization of multi-layer structures, “brick-and-mortar” structures, aerogel structures, asymmetric structures, sandwich structures and other related approaches [20,21,22]. For example, Li et al. fabricated flexible and tough nanofibrillated cellulose/Fe_3_O_4_ and carbon nanotube/polyethylene films with multilayer alternating structures using an alternate vacuum-assisted filtration technique, which demonstrated an electromagnetic interference shielding effectiveness (EMI SE) value of 30.3 dB [23]. Inspired by the structure of “brick and mortar”, Gong et al. fabricated a multifunctional flexible composite membrane composed of PCC/MXene/polyvinyl alcohol (PMP) using a one-step vacuum-assisted filtration method, exhibiting an EMI SE value of 43.13 dB [24]. Aside from that, Fu et al. proposed a laminated structural engineering strategy for the fabrication of an autonomous carbon nanotube-based aerogel film, which exhibited a compacted porous structure that contributed to an enhanced EMI SE value of 35.1 dB through effective internal reflection loss [17]. Among the aforementioned methods, sandwich-structured EMI shielding films can establish conductive networks by concentrating electrically and magnetically conductive fillers in specific layers. Therefore, the sandwich structure not only effectively enhances EMI shielding efficiency but also provides structural support to the entire film and safeguards the intermediate shielding filler against friction, oxidation, and other forms of damage. As they reported, Yao et al. prepared sandwich-structured Ti_3_C_2_T_x_ MXene/ANF films with an EMI SE greater than 49.7 dB by using a layer-by-layer construction method [25]. However, at present, the preparation of sandwich-structured films through vacuum filtration is highly inefficient, necessitating the urgent development of a rapid and large-scale film fabrication method [26].

The incorporation of a high-performance polymer with exceptional mechanical properties and an effective shielding filler represents a viable approach to enhance the mechanical characteristics of EMI shielding composite films [27,28,29]. Aramid fiber is such a kind of high-performance polymer fiber which has been widely studied and developed in recent decades [30,31,32,33]. It is known for its excellent strength, high modulus, and high heat resistance [34], and since Takayanagi et al. discovered the solubility of aramid, researchers have found a simple preparation method for aramid nanofibers (ANFs) [35]. ANFs are extracted from macro aramid fibers by chemical etching and stripping, which is quite simple and fast at present [36,37,38]. ANFs inherit the excellent mechanical properties and thermal stability of macro aramid fibers. Wang et al. constructed a bidirectional conductive network to prepare a dual-function thermal management material. The film had a thermal conductivity of 31.3 W/mK and a mechanical strength of more than 100 MPa [39]. At the same time, at the nano and micro level, it can be easily prepared into a film, which has become a basic building unit of high-performance composite materials and attracted our strong attention. For example, Wang et al. added poly(diallyldimethylammonium chloride)-functionalized nanodiamond (ND@PDDA) to ANF/DMSO blending and scraped protonation film, which is of great significance for practical engineering applications [40]. Zhou et al. prepared ANF@PPy thin films with an EMI SE of 41.69 dB when the amount of pyrrole (Py) monomer was 40 uL [41]. In addition, among various EMI shielding fillers, silver nanowires with a one-dimensional structure and high aspect ratio have ultra-high electrical conductivity up to 6.3 × 10^7^ S/m. Therefore, it is easy to construct excellent two-dimensional conductive network structures and obtain excellent flexibility, which can be widely used in EMI shielding films. For example, Zeng et al. prepared WPU/AgNW nanocomposites with unidirectionally aligned micrometer-sized pores, where only 28.6 wt% of the AgNWs could reach up to 64 dB in the X band [42].

In this work, the ANFs were prepared by dissolving aramid fibers in an alkaline solution of dimethyl sulfoxide using the aramid deprotonation method. Subsequently, an NM was utilized as a scaffold onto which the ANF solution was applied and subsequently water-bathed, resulting in the formation of an aramid nanofiber film (ANF) supported by the scaffold. To create a conductive network structure with excellent EMI shielding properties, an AgNW solution was sprayed onto the surface of the film. Furthermore, to form a sandwich structure, another coating scraping step with the ANF solution was repeated to encapsulate the AgNW conductive network structure inside. This sandwich-structured film allows for controllable the EMI shielding efficiency by adjusting the loading amount of AgNWs. Additionally, due to its inner NM skeleton and outer ANF coatings, this film exhibits exceptional mechanical properties and thermal stability. Our work holds significant implications for facilitating rapid industrialized production of high-performance EMI shielding films.

## 2. Experimental Section

### 2.1. Chemicals and Materials

Poly-paraphenylene terephthalamide (PPTA) was bought from Yantai Tayho Advanced Materials Group Co., Ltd. in Yantai, China. The nylon mesh (NM, 200 mesh) was provided by Changzhou Hongli Hardware Co., Ltd. in Changzhou, China. Dimethyl sulfoxide (DMSO), potassium hydroxide (KOH), polyvinyl pyrrolidone (PVP, Mw ≈ 5.8000), and glycerol were all purchased from Shanghai Aladdin Biochemical Technology Co., Ltd. in Shanghai, China. Silver nitrate (AgNO_3_) was supplied by Guangdong Guanghua Sci-Tech Co., Ltd. in Shantou, China. Sodium chloride (NaCl) was supplied by Tianjin Baishi Chemical Industry Co., Ltd. in Tianjin, China. Deionized water (DI water) was supplied by Minling Material in Hangzhou, China. All chemicals were used without further purification.

### 2.2. Preparation of AgNWs

The AgNWs were synthesized using the well-established polyol reduction method [43,44], and 190 mL of glycerol and 5.86 g of PVP were added to a 250 mL three-necked flask, followed by gradual heating from room temperature to 80 °C at a lower rotational speed. The temperature was then maintained between 80 and 90 °C until complete dissolution of the PVP was achieved. Subsequently, the solution was allowed to cool down after discontinuing the heating process. Meanwhile, a mixture of 0.059 g of NaCl in 500 μL of DI water and 10 mL of glycerol was prepared by homogeneous mixing and preheated at 60 °C for at least 5 min. Once the PVP-propanetriol solution reached a temperature of 55 °C, it was supplemented with 1.58 g of AgNO_3_ followed by the addition of the preheated NaCl mixture. The whole reaction system was gradually heated up to 210 °C with a lower stirring speed, and then the heating was stopped. The gray-green product was taken out and cooled to room temperature, and then a large amount of DI water was added before it was allowed to stand for one week. After that, the mixture underwent three rounds of centrifugation at 4000 rpm, using DI water as the washing agent. Ultimately, the AgNWs were dispersed in ethanol to yield an AgNW solution.

### 2.3. Preparation of ANF/DMSO Solution

The ANFs were prepared with the previously reported deprotonation process [35,45,46]. Specifically, 1 g of KOH was dissolved in a mixed solvent containing 2 mL of DI water and 100 mL of DMSO. Then, 1 g of PPTA was added. After stirring for 4 h at room temperature, a dark red viscous ANF/DMSO solution was obtained with a concentration of 10 mg/mL.

### 2.4. Preparation of NAAANF Films

The preparation process of the NAAANF is illustrated in Figure 1a. The entire experiment was carried out at room temperature. Firstly, the ANF/DMSO solution was coated on the transparent NM (Figure 1b), and then the complete film was immediately immersed in water for 5 min to ensure thorough protonation of the ANFs. In this process, the reddish brown film would gradually become transparent, which means that ANF protonation was complete and formed a cross-linked ANF. Subsequently, the wet film was dried at 60 °C for 30 min in an oven, resulting in the formation of NANF films (Figure 1c). The inner EMI shielding layer was prepared by spraying the required amount of AgNWs on the NANF surface, and the film was denoted as NAANF (Figure 1d). It has a silvery metallic luster. Finally, the above steps were repeated to prepare another layer of ANF on the AgNW coating to obtain an NAAANF EMI shielding film (Figure 1e). The ultimate specimen exhibits a light gray appearance. These films were labeled as NAAANF0.2, NAAANF0.4, NAAANF0.6, NAAANF0.8, and NAAANF1.0 based on the amount of AgNWs added (ranging from 0.2 to 1.0 mg/cm^2^).

### 2.5. Characterizations

The morphology and microstructure of the AgNWs, ANFs, NANF, NAANF, and NAAANF were observed through field emission scanning electron microscopy (SEM, GeminiSEM500, Zeiss, Jena, Germany). The sample was sprayed with gold for 5 min prior to testing, and the acceleration voltage during the test was 3 kV. The microscopic morphology of the raw materials of the ANFs and AgNWs was observed using transmission electron microscopy (TEM, JEM-1400Flash, JEOL, Akishima-shi, Japan) at an acceleration voltage of 120 kV. X-ray diffraction (XRD, D8 Advance, Bruker-AXS, Billerica, MA, USA) of the sample used a copper target with diffraction angles ranging from 5 to 90°. Thermogravimetric analysis (TGA, TG209F1, NETZSCH, Selbu, Germany) was used to analyze the thermal stability of the samples. The analysis was conducted in a nitrogen atmosphere, with a temperature range from 25 to 800 °C and a heating rate of 10 °C/min. A four-finger probe tester (FT-340, CHNT, Leqing, China) was used to measure the resistance of different AgNW concentrations. The EMI SE properties of the composite in the 8–12 GHz (X-band) microwave range were measured with the waveguide method using a vector network analyzer (VNA, AV3672, Ceyear, Qingdao, China). The measured scattering parameters (S11 and S21) were used to calculate the EMI SE of the materials, from which the total EMI SE (SE_T_), absorbing shielding effectiveness (SE_A_), reflecting shielding effectiveness (SE_R_), and power coefficients of absorptivity (A), reflectivity (R), and transmittance (T) were calculated as follows:R = |S11|^2^
T = |S21|^2^
1 = A + R + T
SE_R_ = −10 log (1 − R)
SE_A_ = −10 log (T/(1 − R)
SE_T_ = SE_A_ + SE_R_ + SE_M_

Multiple reflection (SE_M_) was generally ignored when SE_T_ > 15 dB [47]. A tensile stress and strain test was performed on a universal testing machine (KJ-1065B, Kejian, Dongguan, China) at a speed of 10 mm/min to test the strip sample of 10 cm × 1 cm [48].

## 3. Results and Discussion

### 3.1. Morphologies and Microstructures of ANFs and AgNWs

The purchased PPTA (Figure 1a) was dissolved in a KOH/DMSO solution to obtain an ANF/DMSO solution (Figure 1b), which was then added to DI water and dispersed using ultrasonication to obtain an ANF dispersion solution (Figure 1c). In order to confirm the successful preparation of ANFs, the ANFs were characterized by SEM (Figure 1d,e) and TEM (Figure 1f). The SEM image revealed the one-dimensional fiber-like structure of the ANFs, with diameters ranging from 20 to 30 nm. Additionally, Figure 1g is a comparison of the XRD patterns of the ANFs and PPTA. Diffraction peaks at 20.6°, 23.0°, 28.7°, and 39.1° appeared for the PPTA, which corresponded to the (110), (200), (004), and (006) crystal faces of the PPTA. Meanwhile, the XRD spectra of the ANFs were almost flattened, which suggests that the crystal structure of the PPTA was disrupted by deprotonation of the ANFs, further indicating the formation of ANFs. To assess whether the thermodynamic properties were retained during the conversion from PPTA to ANFs, TGA (Figure 1h) and DTG (Figure 1i) curves were performed under a nitrogen atmosphere. Specific data can be seen in Table 1. In the first stage of weightlessness, the PPTA and ANFs lost water and organic solvents in the material to varying degrees. After a period of stability, the ANFs became weightless at 560 °C, followed by the PPTA at 598 °C. The results indicate that both the PPTA and ANFs exhibited weight loss at temperatures above 500 °C, suggesting that the thermal stability of the ANFs was slightly lower than that of the PPTA, but it still had good performance.

The AgNWs were simply synthesized by a polyol reduction method and uniformly dispersed in anhydrous ethanol (Figure 2a). The AgNW dispersion liquid exhibited a silver-gray color with a slight green tint. SEM analysis revealed the microstructure of the prepared AgNWs, as shown in Figure 2b, and TEM analysis further confirmed their characteristics, as depicted in Figure 2c. It was observed that the length of the AgNWs was approximately 20 μm, with a diameter of less than 100 nm and a length-to-length ratio exceeding 200. Some slightly curved AgNWs observed through SEM indicated their inherent flexibility, making them suitable for applications in flexible film materials. Diffraction peaks at 38.1°, 44.2°, 64.3°, and 77.4° appeared in the XRD (Figure 2d) patterns. The peaks could be designated as (111), (200), (220), and (311) facets, respectively¸ all of which were in excellent accordance with the faced-centered cubic (FCC) Ag crystal nanostructure.

### 3.2. Structural Characterization of NAAANF

Figure 1a illustrates the fabrication process of the NAAANF composite films. The colorless and translucent 200 mesh NM yarn with warp and weft interlacing (Figure 3a1) demonstrated a certain level of mechanical strength. Owing to the utilization of NM as a foundation, the ANF/DMSO solution was meticulously coated onto the NM via a blade process, followed by protonation to generate a functional membrane. The SEM image of the NANF is shown in Figure 3b. It appeared to be light yellow and semi-transparent (Figure 1c), resembling a leaf, where the NM acted as the leaf veins and the ANF acted as the leaf blade. Subsequently, various amounts of AgNWs were sprayed onto the NANF surface to form a conductive network structure. The NAANF had the same metallic luster as the AgNWs we synthesized (Figure 1d). SEM images of the NAANF with different AgNW concentrations are shown in Figure 3c–g. NAANF0.2 (Figure 3c), containing 0.2 mg/cm^2^ of AgNWs, exhibited significant spacing in the conductive network, which became tighter as the content of the AgNWs increased. The saturation point in particular was reached at 0.8 mg/cm^2^ of AgNWs deposited by spraying on the NANF, which was particularly evident in NAANF1.0. Finally, to protect the exposed AgNW conductive network, the scraping and protonation steps described above were repeated for the NAANF. The AgNWs were completely enveloped by the final layer of ANF, resulting in the formation of a sandwich structure composed of ANF-AgNWs-ANF. The preparation of NAAANF was successfully completed. The final NAAANF was in a grayish-yellow film state (Figure 1e). The SEM image in Figure 3h illustrates the encapsulation of AgNWs by the final layer of the ANF film, providing effective protection against detachment and oxidation.

### 3.3. Electrical Conductivity and EMI Shielding Properties of NAAANF

Electric and magnetic fields interact with each other to form EM waves, which propagate in a manner similar to sound waves. These waves undergo changes when they transition from one medium to another. EM waves encounter impedance mismatches and produced numerous reflections when interacting with conductive materials in different media. Materials with better conductivity are more likely to cause impedance mismatches, and thus their shielding performance is relatively superior. In general, the better the conductivity of the material measured, the higher the EMI SE value. Figure 4a shows the electrical conductivity of the NAAANF films with different loading amounts of AgNWs. As the AgNW content increased, the electrical conductivity of the film increased, with the NAAANF1.0 sample exhibiting an electrical conductivity of 123.7 S/cm. Additionally, Figure 4b depicts the construction of a conductive channel for a small light bulb, which visualizes the different conductive properties of the NAAANF films. When an applied voltage of 3 V was utilized, the brightness of the light bulb was found to increase with the concentration of AgNWs in the film.

Due to the high conductivity of the AgNW coating in NAAANF films, significant impedance mismatches and substantial conduction losses occurred when exposed to EM waves, making them highly efficient in EMI shielding. To evaluate their performance, we evaluated the EMI SE in the frequency range of 8.2–12.4 GHz (X band). As anticipated, a direct correlation between the conductivity trends and the EMI SE enhancements of the NAAANF films was observed, as depicted in Figure 4c. Remarkably, the NAAANF0.2 sample displayed an average EMI SE of 28.7 dB, indicating its ability to attenuate 99.884% of incident EM waves, surpassing the requirements for commercial EMI shielding (20 dB). Subsequently, the NAAANF0.4, NAAANF0.6, NAAANF0.8, and NAAANF1.0 samples exhibited progressively increasing average EMI SE values of 37.7 dB, 46.4 dB, 49.5 dB, and 50.6 dB, respectively. Notably, the EMI SE improvement from NAAANF0.8 to NAAANF1.0 was marginal, suggesting that the EMI shielding effectiveness reached a saturation point at an AgNW concentration of 0.8 mg/cm^2^. This observation further substantiated our previous analysis, indicating that the maximum EMI shielding performance was attained when the conductive plane of the NAAANF film was fully covered and reached saturation.

The shielding mechanisms of EMI shielding materials include reflection, absorption, and multiple reflection. To elucidate the EMI shielding mechanism of NAAANF, we calculated the EMI shielding effects in terms of EMI SE_T_, EMI SE_R_, and EMI SE_A_ as well as the corresponding coefficients T, R, and A, as shown in Figure 4d,e. Nevertheless, the basic principle of EM wave propagation had to be taken into account. When an EM wave encountered an EMI shielding material in an air medium, an initial impedance mismatch triggered a large reflection, which then partially penetrated the material and was subsequently absorbed through dielectric losses. It is noteworthy that the SE_A_ values for the NAAANF series surpassed their respective SE_R_ values. Did this suggest that NAAANF had potent absorption properties concerning EMI shielding? Considering the fundamental principle of EMI shielding, when an EM wave met an EMI shielding material in an air medium, an initial impedance mismatch caused a substantial reflection. Subsequently, part of the wave penetrated the material and underwent absorption due to dielectric loss. An analysis of the SE_R_ values across the entire film revealed that over 90% of the incident EM wave was efficiently reflected upon interacting with NAAANF. Furthermore, coefficients R and A offered additional insights. Notably, the R value significantly outpaced the A value, reinforcing the notion that the NAAANF series primarily functioned as a reflective EMI shielding material.

NAAANF films are often exposed to complex and harsh environmental climates during actual use. The most important thing was that the upper and lower layers of ANF protected the AgNW conductive layer. We needed to conduct a test in an extreme environment after treatment of the NAAANF film to see if it could maintain better EMI shielding performance. As shown in Figure 4f, sample NAAANF0.4 had been soaked in an acidic solution (pH = 2), alkaline solution (pH = 13), and DI water (pH = 7) for 48 h and was then dried. The curves of the samples treated under the three conditions fluctuated slightly, but their EMI shielding performance remained quite good, with average EMI SE values of 38.6, 38.7, and 40.2 dB, respectively. This fully demonstrated that the ANF film was resistant to acid and alkali and effectively protected the conductive network structure of the AgNWs from damage. The experiment proved that the EMI film had excellent environmental performance. Figure 4g showed the thickness and EMI shielding performance of this work compared with other recent studies. The sample in this work achieved quite excellent EMI shielding performance under extremely thin conditions, allowing it to be applied in more extreme environments. Figure 4h shows the structure and EMI shielding mechanism of the NAAANF. A high impedance mismatch occurred when the incident EM wave reached the surface of the thin film and the highly conductive network composed of AgNWs. Therefore, 90% of the EM wave was reflected, a small amount of the EM wave was polarized and dissipated as heat in the conductive network, and a small amount of the EM wave passed through the film.

### 3.4. Mechanical Properties and Thermal Stability of NAAANF

Basic mechanical properties needed to be taken into account for thin film material applications. The NM used in this film had good basic mechanical properties, and it was expected that the scratch coating addition of ANF would further enhance the mechanical properties of the NAAANF. Figure 5a,b illustrate typical stress–strain plots for the NM, NANF, and NAAANF films along with the corresponding tensile strength and elongation. As the NM served as the supporting framework for the film, during external tensile fracturing, certain yarns within the NM underwent initial rupturing due to uneven stress distribution. For the stress–strain curve, this characteristic was reflected as a step-wise fracture behavior in the deformation stage of the NM and NANF. In terms of mechanical strength, NANF demonstrated a slight enhancement compared with the NM, with the tensile strength incrementing from 35.7 MPa to 38.6 MPa and the strain increase changing from 85.4% to 75.5% at the point of fracture initiation. However, owing to the thicker ANF coating resulting from the deposition of two ANFs, the NAAANF films exhibited increased interconnectivity among the ANFs, which significantly affected the manifestation of the stress–strain curve for the thin film. Notably, the NAAANF films displayed a single distinct fracture point, in contrast to the multiple fracture points observed in the NM and NANF, indicative of the uniform distribution of external forces facilitated by the ANFs. Among the NAAANF series samples, NAAANF0.2 and NAAANF0.4 demonstrated favorable tensile strength, reaching 50 MPa. NAAANF0.6 showed a slight decline, while NAAANF0.8 and NAAANF1.0 exhibited tensile strengths of approximately 40 MPa, similar to that of the NANF. Overall, the strain level for the NAAANF resided around 29%, highlighting its superior overall structural integrity compared with the NM and NANF.

The samples were immersed in acidic, alkaline, and neutral solutions for 48 h to evaluate their mechanical properties. The corresponding data can be observed in Figure 5c,d, revealing that the film’s mechanical properties remained largely unaffected following exposure to extreme environmental conditions. The stress–strain characteristics fell within the margin of error, indicating that the simple scraping coating of an ANF film effectively safeguarded it against degradation caused by substances under extreme conditions. Moreover, this ANF film not only shielded the NM yarn but also provided protection for the AgNWs. Thus, the NAAANF film demonstrated remarkable resistance to environmental factors.

Figure 5e,f depicts the TGA and DTG curves, respectively, for the NM, NANF, and NAAANF under a nitrogen atmosphere. Additionally, Table 2 presents the corresponding data in detail. We observed that the initial curves of the NM, NANF, and NAAANF samples were identical, with weight loss commencing at temperatures exceeding 390 **°C**. This stage signifies a significant reduction in weight, attributed to the pyrolysis of the NM. Subsequently, the NANF and NAAANF exhibited lesser weight loss around 560 **°C** due to the presence of the ANF. This observation aligns with the known pyrolysis temperature range for ANFs. Lastly, the residual carbon content in the NANF and NAAANF slightly surpassed that of the NM, owing to the inclusion of ANF and AgNWs.

## 4. Conclusions

Based on the aforementioned statement, we successfully fabricated a flexible, high-performance EMI shielding film that offered rapid preparation and adaptability to various environmental conditions. We used a scratch-coating ANF and spray-coating AgNWs method to prepare sandwich-structured NAAANF films. This method is cleverly designed and allows for the rapid preparation of ultra-thin electromagnetic shielding films. The addition of 1.0 mg/cm^2^ AgNWs resulted in a shielding effectiveness of up to 50.6 dB. Furthermore, the film was subjected to extreme acidic and alkaline environments, yet its electromagnetic shielding performance remained excellent. Our research contributes novel insights to the domain of ultra-thin, high-performance EMI shielding films. The simplicity of operation and cost-effectiveness make this film particularly promising for widespread utilization in the realm of electronic devices.

## Data Availability

The data presented in this study are available in the article.

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
