# Peer review of "Large-Scale Fabrication of Tunable Sandwich-Structured Silver Nanowires and Aramid Nanofiber Films for Exceptional Electromagnetic Interference (EMI) Shielding"

_polymers, 2023, doi:10.3390/polym16010061_

Round 1
Reviewer 1 Report
Comments and Suggestions for Authors
This manuscript discusses the large-scale fabrication of polymer composite with Exceptional Electromagnetic Interference (EMI) Shielding. The composite has a nylon mesh (NM) backbone, with AgNWs serving as the shielding coating and aramid nanofibers (ANFs) acting as the cladding layer.
Reviewer comment.
· The abstract needs to be rewritten and more results need tot be added to it.
· Line 21 Advancement not Advancements
· for example. Line 31-33 “ Due to the outstanding thermal resistance, acid and alkali resistance properties of aramid fibers, the NAAANF film demonstrates remarkable robustness in the face of complex usage environments”
· can be written “ The NAAANF film demonstrated remarkable robustness in the face of complex usage environments as a result of the outstanding thermal, acid and alkali resistance properties of aramid fibers.
· Line 38-48. Rewrite please, you talked about the same idea over and over “lightweight, flexible”.
· Line 84-85 it is thermal conductivity not thermal stability.
· Check the English language
· Define the following
1. ND
2. PDDA
3. ANF
4. DMSO
· I can’t see your novelty, how your work is novel? What is the difference between what other did and you do?
· Why the Preparation of AgNWs is written in a deferent font and bold.
· could you please come up with different shorter names than NANF, NAANF and NAAANF. As it is confusing and hard to follow.
· Line 93-94 I believe (figure 1 e) is SEM also. however, (figure 1 f) is TEM.
· I can’t find any XRD in figure 1. Please modify your figure and your text along with discussion.
· I can’t see any discussion for RAMAN and TGA!!!!!!!!!!
· I believe to you need to think about Figure 3. SEM image of the NM. Do you really need to provide all these SEM images, I am not sure what you are trying to prove here!
· Figure 4 c, d, e, fand g are so small. Please provide better figures
· It would be better if you provide electrical conductivity not the electrical resistance for figure 4 a
· Could you explain more the relation between electrical conductivity and the EMI SH
· Line 299-300 you stated that “An analysis of the SER values across the entire film revealed that over 90% of the incident EM wave was efficiently reflected upon interacting with NAAANF”. However, Figure 4 f shows that you have an absorption dominant mechanism. Could you please explain more about this?
· It would strengthen your results if you can also provide mechanical results for your composite after the acid, alkaline and water 48 hrs. exposure.
· Expand your conclusion and add some more results.
Comments on the Quality of English Language
You need to proofread
Reviewer 2 Report
Comments and Suggestions for Authors
The manuscript by Xinbo Jiang et al. is considering the problem of fabricating thin composite films for the aim of the electromagnetic interference shielding. For this the authors propose to use silver nanowires and aramid nanofibers on a nylon mesh backbone. The main novelty of the work is the preparation of the films which are grown from a solution rather than produced by stacking and pressurizing layers in vacuum. This potentially allows for higher efficiency of the production.
I found the paper to be well-written and scientifically sound. I also failed to find the papers reporting the same idea before. However, I would suggest to add a couple of following citations on the papers investigating aramid nanofiber/silver nanowire films for EMI shielding:
1. Shuang Li et al. Composites Part A 151, 106643 (2021) https://doi.org/10.1016/j.compositesa.2021.106643
2. Dan Guo et al. Journal of Alloys and Compounds 923, 166401 (2022) https://doi.org/10.1016/j.jallcom.2022.166401
It would be also helpful to add some comparison of the properties of the films investigated in the current paper with the results of those articles as well as with the one reported in [43]. Particularly, it is important to understand if there are any disadvantages of the authors' film production method comparing to the layered one. For example, I guess that the films grown from solution have lower mechanical strength.
Author Response
Dear reviewer,
Thank you very much for your kindly comments on our manuscript. There is no doubt that these comments are valuable and very helpful for revising and improving our manuscript. In what follows, we would like to answer the questions you mentioned and give detailed account of the changes made to the original manuscript.
The manuscript by Xinbo Jiang et al. is considering the problem of fabricating thin composite films for the aim of the electromagnetic interference shielding. For this the authors propose to use silver nanowires and aramid nanofibers on a nylon mesh backbone. The main novelty of the work is the preparation of the films which are grown from a solution rather than produced by stacking and pressurizing layers in vacuum. This potentially allows for higher efficiency of the production.
I found the paper to be well-written and scientifically sound. I also failed to find the papers reporting the same idea before. However, I would suggest to add a couple of following citations on the papers investigating aramid nanofiber/silver nanowire films for EMI shielding:
- Shuang Li et al. Composites Part A 151, 106643 (2021) https://doi.org/10.1016/j.compositesa.2021.106643
- Dan Guo et al. Journal of Alloys and Compounds 923, 166401 (2022) https://doi.org/10.1016/j.jallcom.2022.166401
Reply: Thank you for your suggestion. We have added the relevant literature.
It would be also helpful to add some comparison of the properties of the films investigated in the current paper with the results of those articles as well as with the one reported in [43]. Particularly, it is important to understand if there are any disadvantages of the authors' film production method comparing to the layered one. For example, I guess that the films grown from solution have lower mechanical strength.
Reply: Thank you for your question. As mentioned above, our innovative idea is different from the extraction filter film [43]. Your point is correct. The film they filter has better mechanical strength, but rapid scale production is our biggest advantage. In order to take into account the mechanical properties of the film, we use nylon mesh is the overall mechanical support of the film.
Thank you again for your positive and constructive comments and suggestions on our manuscript. We hope you will find our revised manuscript acceptable for publication.
Round 2
Reviewer 1 Report
Comments and Suggestions for Authors
accepted